Impact of historical land use change on the brown bear habitat connectivity in the Polish Carpathians

Szwagierczak Anna anna.szwagierczak@student.uj.edu.pl 1
Ziółkowska El.zbieta 1 2 3
Wąs Joanna 4
Jakiel Michał 1
Kaim Dominik 1
1 Institute of Geography and Spatial Management, Faculty of Geography and Geology, Jagiellonian University , Kraków , Poland
2 Institute of Environmental Sciences, Faculty of Biology, Jagiellonian University , Kraków , Poland
3 Social-Ecological Systems Simulation Centre, Department of Agroecology–Agricultural Biodiversity, Aarhus University , Aarhus , Denmark
4 Department of Geoenvironmental Research, Institute of Geography and Spatial Organization, Polish Academy of Sciences , Kraków , Poland
Phairuang Worradorn
Electronic publication date: 2025 Nov 11
Publication date: 2025
Volume: 13
Electronic Location ID: e20295
Received 2025 May 13; Accepted 2025 Oct 6
Copyright: ©2025 Szwagierczak et al.
Copyright year: 2025
Copyright holder: Szwagierczak et al.
License: This is an open access article distributed under the terms of the Creative Commons Attribution License, which permits unrestricted use, distribution, reproduction and adaptation in any medium and for any purpose provided that it is properly attributed. For attribution, the original author(s), title, publication source (PeerJ) and either DOI or URL of the article must be cited.
License URL: https://creativecommons.org/licenses/by/4.0/

Keywords: Land use change, Depopulation, Habitat connectivity, Brown bear, Carpathian Mountains, Historical maps

Funding: CELSA Research Fund within the project ‘The impact of depopulation on ecosystem services in Europe. A pilot study in France, Czech Republic and Poland’ no 3E22062 The research was supported by the CELSA Research Fund within the project ‘The impact of depopulation on ecosystem services in Europe. A pilot study in France, Czech Republic and Poland’, no 3E220627. The funders had no role in study design, data collection and analysis, decision to publish, or preparation of the manuscript.

==============================
Background

Europe has undergone dynamic land use changes in recent decades that have affected the extent, quality, and connectivity of large carnivore habitats. However, the current distribution of large carnivores also depends on historical land use processes. In this article, we analyse the impact of historical land use changes on the potential connectivity of brown bear habitats in the region linking the western and eastern parts of the Carpathians, one of Europe’s biodiversity hotspots.

Methods

The analyses were conducted based on elevation, slope, and distance-based, land use-related variables representing four time periods: 1860s, 1930s, 1970s, and 2013, using cost surface and least-cost path analyses. We used two different approaches to create cost surfaces: weighted, where the weights differentiated between variables according to their relative importance, reflecting their role in either bear space selection or avoidance, and unweighted, where all the variables were treated as equally important.

Results

The results of both approaches showed a gradual improvement in habitat connectivity for brown bears over time, driven by the increase in forest cover observed over the whole analysed period. However, the dynamics of these changes were much higher after the forced post-war resettlement in the 1940s. These tragic events resulted in the removal of settlements over large areas, substantially reducing human pressure and allowing brown bears to spread into new territories, expanding their habitats and creating new connectivity opportunities. We found that up to 40% of the current corridor was stable since mid-19th century. Our analysis shows that the current population decline in many rural areas of Europe may have positive implications for the habitats and population connectivity of large carnivores, but careful planning is needed to avoid negative interactions with local communities.

Introduction

Land use change has affected almost one third of the global land area since the 1960s (Winkler et al., 2021), substantially influencing the connectivity of mammalian habitats and their conservation (Di Minin et al., 2016). The lack of habitat connectivity has been recognized as one of the most critical threats to species, as it limits the likelihood of dispersal and genetic exchange (Ripple et al., 2014). These processes are essential for mitigating the negative effects of climate change on species (Schloss, Nuñez & Lawler, 2012; Sıkdokur et al et al., 2025) and reducing the extinction risk (Davis, Faurby & Svenning, 2018).

Changes in land use, and consequently habitat connectivity, are not, however, uniform globally. Even within Europe itself, vastly different trends in land use change can be observed, as landscapes with highly fragmented habitats (Ibisch et al., 2016) co-occur with regions experiencing land abandonment on a large scale (Navarro & Pereira, 2015; Lasanta et al., 2017; Ustaoglu & Collier, 2018; Kolecka, 2021). Processes of land abandonment are often followed by forest cover increase, which have been observed for decades in many, especially remote, areas of the continent (Kaim et al., 2016; Loran, Ginzler & Bürgi, 2016; Abadie et al., 2018; Lieskovský et al., 2018). These abandoned areas, where human pressure has decreased and the availability of shelter and prey has increased, may provide unique opportunities for rewilding (Navarro & Pereira, 2015; Araújo & Alagador, 2024), which can also support further the recovery of large carnivores in Europe (Chapron et al., 2014; Cimatti et al., 2021; Bernardi et al., 2025).

Although the importance of incorporating land use and human population changes over time has been recognized in species distribution and habitat suitability studies (Conlisk et al., 2013), temporal dynamics are rarely considered in habitat connectivity analyses which typically focus on animal utilisation of the current land use (Anijalg et al., 2020). Nevertheless, in most cases there exist a clear time lag between the change occurring in land use and the utilisation of the connectivity followed by that change (Auffret, Plue & Cousins, 2015; Chen et al., 2023). That is why it is important to take into account the past land use change and its legacies in conservation, restoration or biodiversity monitoring (Reinula et al., 2024). Considering the long-term effects of land use changes on species is essential for setting reliable recovery and restoration targets and identifying key opportunities where conservation actions can be most effective (Grace et al., 2019; Clavero et al., 2023).

The Carpathian Mountains are one of Europe’s biodiversity hotspots, supporting a diverse range of large carnivores and herbivores (Kozak et al., 2013). At the same time, the region has experienced substantial and dynamic land use changes (Munteanu et al., 2014; Lieskovský et al., 2018). Political decisions have led to large-scale population displacements in some parts of the area, resulting in a substantial increase in forest cover and an overall decrease in human pressure (Bičík, Jeleček & Štěpánek, 2001; Yin et al., 2019; Affek et al., 2021). While greater habitat availability does not automatically translate to improved ecological connectivity due to potential movement barriers, these landscape changes could potentially facilitate connectivity for large carnivores. This is especially important in European landscapes, where habitats are highly fragmented (Ibisch et al., 2016), substantially limiting the recolonisation of large species (Zedrosser & Swenson, 2023; Bluhm et al., 2023).

One species that could benefit from these land use changes is the brown bear (Ursus arctos), the largest terrestrial carnivore currently living in Europe (Fernández et al., 2012). By the end of the 19th century, the Carpathian brown bear population in the northern part of the region had become isolated from the Alpine population, and then, by the end of the First World War, further subdivided into western and eastern groups within the Carpathians (Hartl & Hell, 1994), which is currently visible in the subpopulation diversity (Straka et al., 2012; Matosiuk et al., 2019; Lucas et al., 2025). Although bears were sporadically observed in various parts of the Polish Carpathians during the inter-war period (Niezabitowski, 1933; Jakubiec, 2001), a substantial part of the population lived in the eastern part of the mountains, in present-day Ukraine (Niezabitowski, 1933). Post-war resettlements in the eastern Polish Carpathians in the 1940s, followed by a substantial reduction in human activity and consequent increase in forest cover, provided a unique opportunity for bear recovery. This led to an increase in bear numbers and an expansion of their range, with occasional sightings of dispersing individuals outside the mountains (Jakubiec, 2001). Post-war forced resettlements facilitated the reconnection of western and eastern bear habitats in the Polish Carpathians through the Beskid Niski Mountains (Jakubiec & Buchalczyk, 1987). However, the process of restoring connectivity and increasing bear habitat availability in this area over time remains poorly understood. Nevertheless, the recent bear occurrence data still indicates that this region is critical for connecting the western and eastern bear habitats in the Polish Carpathians (Kaczensky et al., 2021).

In this paper, we analyse a series of historical land use reconstructions spanning 160 years to assess the changes in bear habitat connectivity in the Beskid Niski Mountains (i.e., between the Western and Eastern Carpathians) in the context of post-war displacement. We aim to answer the following questions:

(1) How has the potential brown bear habitat connectivity between the eastern and western Carpathians changed over the last 160 years?

(2) To what extent current patterns of brown bear habitat connectivity were shaped by legacy effects of historical land use and land cover changes?

Our intention here is not to define the spatial extent of the current migratory corridor, but to assess the legacy effect of past land use on the current habitat connectivity. This is important, as farmland abandonment and forest succession are not only historically relevant processes but are also continuously observed (Kolecka et al., 2017), understanding the role of these land use changes is crucial for the current and future management of bear populations. What is more, integrating past data in current modelling approaches may shed a new light on the connectivity effectiveness. This is particularly important in the context of human-wildlife interactions, which may increase as a result of habitat recolonisation by brown bears (Chapron et al., 2014; Ziółkowska et al., 2016; Kaczensky et al., 2021; Sıkdokur et al., 2024).

Materials & Methods

Study area

The Beskid Niski Mountains, one of the lowest mountain ranges in the Carpathians, extend over 100 km, with the highest elevations reaching about 1000 m asl. and the main range descending to 500 m asl. The range is relatively narrow and situated between lowlands to the north and south (Figs. 1A, 1B). This makes it a natural link between the Western Carpathians, which span Austria, Czechia, Hungary, Poland and Slovakia, and the Eastern Carpathians, which stretch from Poland and Slovakia through Ukraine towards Romania and Serbia (Fig. 1A). Historically, the region was densely populated by people whose primary activity was agriculture, which shaped the local landscape for centuries until the 1940s. During the 1940s, the local inhabitants, predominantly from the Ukrainian ethnic group of Lemkos, were resettled to the Soviet Union or to western and northern Poland as a result of large-scale forced displacements (Affek et al., 2021). Similar processes substantially altered the neighbouring eastern Carpathian region Bieszczady Mountains. The neighbouring region to the west, the Beskid Sądecki Mountains, was also affected, though to a much lesser extent (Fig. 1B) (Kozak, Estreguil & Troll, 2007; Munteanu et al., 2014).

Figure 1 Location of the study area within Europe and the Carpathians (A) and within the Polish Carpathians (B). Changes in forest cover (1860s–2013) within the study area are shown in (C).

Digital elevation model ©ESRI, USGS, NOAA.

The tragic events of forced population resettlements transformed the area into a natural experiment of rewilding, characterized by rapid forest cover increase and minimal human impact on the landscape over subsequent decades (Kozak et al., 2018; Jabs-Sobocińska et al., 2021; Affek et al., 2021). These changes provided the opportunities initially to enlarge brown bear habitats in the Bieszczady in the east, and subsequently to improve the ecological connectivity between the eastern and western Carpathians via the Beskid Niski area. Although forests now dominate the landscape (Fig. 1C), ongoing land abandonment is visible (Kolecka & Kozak, 2019). Occasionally, signs of recultivation of previously abandoned agricultural land by farmers can be observed (Ortyl & Kasprzyk, 2022).

It is important to add that the post-war resettlements occurred only on the Polish side of the border. The Slovak part of the range, characterised by lower elevation, has experienced a different history and is currently more densely populated and more intensively used for agriculture. Therefore, our study area includes the Beskid Niski Mountains on the Polish side of the border, along with the neighbouring mountain ranges—the Beskid Sądecki to the west and the Bieszczady to the east—which are considered areas of permanent bear presence and between which connectivity was analysed (Kaczensky et al., 2024; Fig. 1B).

Factors affecting brown bear habitat use

Based on previous works dealing with brown bear habitat use in the Carpathians, we identified three types of factors: forest cover, human activities (including buildings and transport infrastructure) and relief (elevation and slope) as key factors influencing brown bear habitat use and habitat connectivity in the area (Koren et al., 2011; Fernández et al., 2012; Ziółkowska et al., 2016). Similar factors, however, were considered also in the other works dealing with the brown bear occurrence (Güthlin et al., 2011; Mateo-Sánchez et al., 2015; Eriksen et al., 2018; Iosif et al., 2020; Mohammadi et al., 2021; Cisneros-Araujo et al., 2021; Sıkdokur et al., 2025). The current information on forest cover and human activities was obtained directly from the digital National Database of Topographic Objects (BDOT10k) at a scale of 1:10,000, representing the conditions in 2013. Historical information was obtained by processing detailed military maps from the 1860s (second military survey 1:28,800) and 1930s (Polish military maps 1:100,000) and topographic maps from the 1970s (Polish topographic maps 1:25,000). We used available digital databases on forest cover and buildings derived from these maps, either through manual vectorisation for the 1860s and 1930s (Kaim et al., 2014; Kaim et al., 2021) or automatic extraction for the 1970s (Ostafin et al., 2017; Szubert, Kaim & Kozak, 2024). As the information on buildings for the 1930s was not available from existing databases, it was extracted by the authors by manually updating the database of buildings for the 1860s based on maps from the 1930s. Available digital databases on main roads (Kaim, Szwagrzyk & Ostafin, 2020) and railways (Kaim et al., 2020) for the 1860s, combined with current data on transport infrastructure from the BDOT10k topographic database, were used to reconstruct the situation for the 1930s and 1970s through manual updates via visual inspection of the 1930s and 1970s maps. Relief information was obtained from the EU Digital Elevation Model (EU-DEM) available through the Copernicus Land Monitoring Service (https://land.copernicus.eu/). All data were converted to the raster format with a spatial resolution of 25 m.

Least-cost modelling

To assess habitat connectivity, we used least-cost analysis, a method that evaluates the impact of the matrix between habitat patches on the dispersal of an organism (e.g., Verbeylen et al., 2003; Etherington & Penelope Holland, 2013). This method utilizes a cost surface, represented as a raster layer, which indicates the movement costs associated with each grid cell of the matrix. Based on the cost surface, routes (i.e., least-cost paths or corridors) with the lowest cumulative resistance between destination locations in a landscape are then calculated as a function of distance travelled and cost incurred (Etherington & Penelope Holland, 2013). We decided to use least-cost analysis, since one general direction of the east–west connectivity is critical in the context of our analysis.

Based on the obtained data on forest cover, human activities and relief, we calculated a set of variables which were then combined into cost surfaces representing costs associated with a potential brown bear movement across the Beskid Niski for each of the analysed time periods (1860s, 1930s, 1970s, and 2013) separately (Table 1). Variables related to forest cover included distance from the forest edge to the forest core (as a measure of forest edge effect) and forest cover persistence (understood as forest presence in previous periods; see Grabska-Szwagrzyk et al., 2024). We assume that bears prefer to move through the interior of forest and older forest stands, as these habitats provide more shelter. Variables related to human activity, considered stressors for bears, included the distance to buildings and the distance to transport infrastructure. We assume that bear travel costs increase with decreasing distance to human activity. Relief-related variables included elevation and slope, with the assumption that bear movement costs increase with higher elevation and steeper slopes (Table 1).

Table 1 The cost values used in the habitat connectivity analysis.

Factor	Value	Movement costs	
		Weighted variant	Unweighted variant	
		1860	1930	1970	2013	1860	1930	1970	2013	
Elevation (m a.s.l.)	0–500	2	2	2	2	5	5	5	5	
500–800	5	5	5	5	13	13	13	13	
800–1,000	8	8	8	8	20	20	20	20	
>1,000	10	10	10	10	25	25	25	25	
Slope (°)	0–3	2	2	2	2	3	3	3	3	
3–30	5	5	5	5	6	6	6	6	
30–47	10	10	10	10	13	13	13	13	
>47	20	20	20	20	25	25	25	25	
Distance to main roads and railways (m)	0–100	2	3	4	5	25	25	25	25	
100–500	1	2	3	3	15	15	15	15	
500–1,000	0	1	1	2	10	10	10	10	
>1,000	0	0	0	0	0	0	0	0	
Distance to buildings (m)	0–50	14	16	19	20	25	25	25	25	
50–100	6	7	9	10	13	13	13	13	
100–500	2	3	5	5	6	6	6	6	
500–1,000	1	1	2	2	3	3	3	3	
>1,000	0	0	0	0	0	0	0	0	
Distance from the forest edge towards the forest core (m)	0–200	15	15	15	15	25	25	25	25	
200–400	10	10	10	10	17	17	17	17	
400–600	5	5	5	5	8	8	8	8	
600–800	3	3	3	3	5	5	5	5	
>800	1	1	1	1	2	2	2	2	
Forest persistence (forest presence in previous periods)	no forest	25	25	25	25	25	25	25	25	
1 period	1	7	7	7	1	7	7	7	
2 periods	–	1	5	5	–	1	1	1	
3 periods	–	–	1	3	–	–	–	–	
4 periods	–	–	–	1	–	–	–	–	

These variables were combined into cost surfaces using a relative quantification method, where relative weights were assigned to each variable based on analysis of existing literature on bear habitat and movement preferences, with particular focus on the Northern Carpathians (Koren et al., 2011; Fernández et al., 2012; Ziółkowska et al., 2016). Since the authors used the factors based on diverse datasets and also hard to be directly compared, we had to make it first more generalised, but comparable. Specifically, we extracted from these works the coefficient estimates of the important variables representing our three group of factors (forest-related, human impact-related and topography-related), transformed them into absolute values and standardized by ranking them into four categories, where the class limit values were the quartiles assessed by using Tukey method (1 –≤Q1, 2 –(Q1, Q2], 3 –(Q2, Q3], 4 –>Q3). Doing so, we received four ranks, which were helpful in defining our own margins of the cost values (see Tables S1, S2). The overall cost values were then calculated as combined values of individual cost components, taking into account assigned weights. Two weighting approaches were applied. In the first approach (hereafter referred to as the unweighted variant), all variables were treated as equally important, sharing the same range of assigned cost values, and their importance did not change over the analysed time period (Table 1). In the second approach (hereafter referred to as the weighted variant), the weights differentiated between variables according to their relative importance, reflecting their role in either bear preference or avoidance. Additionally, in this variant, the weights for human-related variables, other than land use, changed over the analysed time period, based on the assumption that the role of human activities increased over the 160-year period, causing more stressors for the brown bear due to higher levels of technological development over time (e.g., related to higher levels of noise, light, or accessibility) (Table 1). In both weighting schemes, variables related to topography (i.e., slope and aspect) had the same weights assigned. In 1994, a large water reservoir (306 ha) was constructed within the study area. As it can be perceived as a total barrier to bear movement, it was included in the 2013 cost surface with a value of No Data.

For each of the analysed time periods (1860s, 1930s, 1970s, and 2013), two independent cost surface layers were generated based on unweighted and weighted variants, which were then standardised to values ranging from 1 to 100% for easier interpretation. For each of these cost surface layers, we applied least-cost analysis to delineate least-cost corridors (or connectivity zones) linking bear habitat areas in the Beskid Sądecki and Bieszczady through the Beskid Niski Mountains. Least-cost corridors were defined as sets of cells for which the cumulative cost between habitat areas falls below a certain user-defined threshold, set in three options at the 10th, 20th or 30th percentile of the sum of all corridors covering the study area. We characterised corridors using four indicators: (1) the total cost of the corridors relative to their area, indicating the total potential movement effort of brown bears, (2) the share of forest in the corridors, indicating preferred movement conditions, (3) the density of buildings within the corridors, indicating difficulties for bear movement, and (4) the percentage of corridors that remained stable relative to the previous time period, representing the impact of land use change dynamics. All analyses were conducted using ArcGIS Pro 3.2 and the Linkage Mapper software (McRae & Kavanagh, 2011).

Finally, we attempted to compare the resulting corridors with the historical and current data on brown bear occurrence in the study area. Historical data on brown bear occurrence were available only for the 1970s (Jakubiec & Buchalczyk, 1987) at the level of state forest administration regions (http://www.bdl.lasy.gov.pl). For the most recent analysed period, we referred our results to the 10 × 10 km data on permanent and sporadic brown bear occurrence, based on Chapron et al. (2014).

Results

The last 160 years have seen a significant increase in forest cover and a decrease in human activity in the study area, with the most substantial changes occurring between the 1930s and 1970s. The Beskid Niski region experienced the highest increase in forest cover, doubling from 28.6% in the 1860s to 66.3% in 2013, with the higher altitudes of this relatively low mountain range becoming almost completely forested. In the neighbouring regions of Beskid Sądecki and Bieszczady, the increase in forest cover was somewhat smaller than in the Beskid Niski, but still substantial, rising from 39.0% to 69.6% and from 52.2% to 87.2%, respectively (Fig. 1C). The greatest decline in the number of buildings occurred between the 1930s and 1970s, during which many villages disappeared or were reduced to isolated farmsteads. At the same time, selected settlements and towns, mostly in the north-western edge of the study area, experienced gradual growth. The development of built-up areas continued locally in the following period, both in towns and in remote areas.

The above-mentioned changes in land use had a significant impact on the course and characteristics of brown bear movement corridors (Fig. 2; Figs. S1, S2). Regardless of the variant, we observed a decrease in total costs within the corridors over time, with visible stabilisation occurring in the most recent period analysed, between the 1970s and 2013. However, the unweighted variant showed a noticeable overall decrease over time (Fig. 3A; Figs. S3A, S4A). The share of forest cover within the corridors increased for both variants, with the largest increase observed between the 1930s and 1970s. The rate of increase was, however, slower in the unweighted variant (Fig. 3B, Figs. S3B, S4B). The density of buildings within the corridors decreased over time in both variants, apart from 1970s in the option based on 10th percentile (Fig. 3C; Fig. S3C, S4C).

Figure 2 Brown bear corridors connecting Beskid Sądecki in the west and Bieszczady in the east through the Beskid Niski Mts.

Colours represent different variants (unweighted vs. weighted) of cost surfaces and analysed time periods (1860s, 1930s, 1970s and 2013). The same colour patterns are used in Figs. 3A and 4. (For other percentile-based options: see Supplemental Figures). Digital elevation model ©ESRI, USGS, NOAA.

Figure 3 Total costs in corridors related to the area of corridors (per m2) (A), proportion of forest area in corridors (B) and buildings density per km2 (C).

(For other percentile-based options: see Supplemental Figures).

The spatial pattern of the land use changes observed since the mid-19th century has also affected changes in the course of the corridors. The majority of the corridor areas consisted of newly established dispersal areas in 1930s in weighted variant and in the 1970s in unweighted in the options based on the 10th and 20th percentile, while the proportion of parts of the corridors established in earlier periods increased over time in most cases (Fig. 2; Figs. S1, S2). The greatest difference between the weighted and unweighted variants of the corridors was observed in the older periods, where the corridor in the weighted variant showed a very different course in space (Fig. 2; Figs. S1, S2). Both variants were relatively coherent in terms of the location of the most stable dispersal path since the 1970s, regardless the percentile-based option.

Figure 4 Corridors referred to the bear presence data for 1970s and 2010s.

(For other percentile-based options: see Supplemental Figures). Sources for bear occurrence data: 1970s: (Jakubiec & Buchalczyk, 1987), 2010s: (Chapron et al., 2014). Digital elevation model ©ESRI, USGS, NOAA.

Discussion

The selected study area of the Beskid Niski Mountains is unique in that we can observe the effects not only of gradual, selective land use changes caused by socio-economic factors, but also the effects of sudden, far-reaching changes caused by political factors. This, together with the fact that the area is located in the biodiversity hotspot of the Carpathian Mountain range that is home to large carnivores, provides an unusual opportunity to study the long-term effects of past and present land use changes on the persistence of species habitats and their connectivity. In this paper, we analyse 160 years of historical land use reconstructions to assess changes in brown bear habitat connectivity in the Beskid Niski Mountains, focusing on the impact of post-war displacement and answering how habitat connectivity between the Eastern and Western Carpathians has evolved and been affected by depopulation.

While gradual, selective land abandonment linked to post-socialist period has only recently emerged in the region (Kolecka et al., 2017; Ortyl & Kasprzyk, 2022), earlier politically driven depopulation after World War II led to widespread and persistent abandonment of agriculturally marginal areas. Thus, unlike the globally common pattern of temporary abandonment followed by recultivation (Crawford et al., 2022), the Carpathian context reflects a long-term land use transformation with lasting effects on landscape connectivity.

We employed two different ways in accounting for the impact of various determinants on bears’ connectivity, by using weighted and unweighted variants. Since the analysed weighted variants used to generate the cost surfaces differed considerably in terms of weights assigned to factors related to human activity (distance to buildings and transport infrastructure), the differences in the course of bear dispersal corridors in different time periods can be directly linked to changes in human pressure experienced in the study area over the last 160 years. The most significant differences in the corridor courses between the unweighted and weighted variants were observed in the 1860s and 1930s. In these periods, the variants showed significantly different courses for the main corridor branches due to the intensity of the land use at that time, and the scattered settlement all over the area (Kozak, Estreguil & Troll, 2007; Kaim et al., 2021; Affek et al., 2021). In the 1970s, there were no major differences in the course of the corridors between the two analysed variants, regardless the percentile threshold. This period saw a significant decrease in settlement compared to the 1930s, due to the large-scale, post-war resettlements. Thus, in the 1970s, the bear connectivity was mainly shaped by the forest availability, which was more similarly weighted in both variants, while depopulated villages did not constitute a significant barrier. The largest differences between variants occurred in areas that still had considerable settlement at the time. After 1970s, an increase in development was observed, but it was more scattered in nature and located more in the west and north. This led to the changes in the corridor courses visible in 2013, when compared to 1970s. We found also that the corridor was most persistent over time (regardless the variant and the percentile threshold) in the area, where it is quite narrow limiting other relatively safe dispersal opportunities (Fig. 2; Figs. S1, S2). This area co-occurs with the area of the Magura National Park highlighting the role of the area-based nature conservation measures.

Our weighted variant, formulated with a relative quantification method, highlighted the substantial impact of factors related to human activity, mainly settlements. This aligns with the results of studies predicting bear habitat and movement preferences based on presence data. For instance, Ziółkowska et al. (2016) demonstrated that, in addition to elevation range, factors related to the density and distance from settlements and road density had the greatest impact on bear movement, regardless of the model considered. These factors were significantly more influential than forest-related variables. Similarly, in habitat suitability studies, settlements and roads are often indicated as more significant than forests. For example, Mateo-Sánchez et al., (2015) found that settlement density was the most important variable in all best-fit models of habitat selection by brown bears in the Cantabrian Range (SW Spain), surpassing road and forest density. Cisneros-Araujo et al. (2021) also showed that human-related factors are more important than forest in their habitat suitability models for the Cantabrian bears, although the difference in importance was not as pronounced as that found by Ziółkowska et al. (2016). This suggests that for bear movement, the negative impact of development and roads may be even more important. On the other hand, the ‘human footprint’ did not emerge as particularly important in habitat models for bears in Iran (Mohammadi et al., 2021). However, it is unclear from the article how the authors calculated (weighted) the “human footprint”, aside from the fact that several variables were included to distinguish the effects of different kinds of human perturbation (i.e., human population density, human infrastructure, and road network).

Studies on habitat suitability and connectivity often use the density of settlements and/or roads as explanatory variables depicting human impacts. In our study, instead of density-based variables, we used distance-based variables, which provide more robust measures of human impact, especially when relating to objects that are sparsely present. This approach was particularly relevant in our study area, where the density of buildings was very low after World War II, and subsequent development was rather scattered, resulting in zero density values for most of the study area. Additionally, using a variable based on distance rather than density avoids the need to arbitrarily decide or account for different variants of the scale at which density is measured.

The course of the least-cost corridors for the past cannot be easily verified; however, we were able to compare it to independent bear occurrence data from the 1970s and 2010s (Fig. 4). This comparison shows that in the 1970s, the main corridor branch in the eastern part of the study area coincides with areas of certain proximity to the bear dens presence according to Jakubiec & Buchalczyk (1987), specifically areas with vast forested patches and a high percentage of persistent forest. The main corridor branch narrows when crossing areas of lower probability of bear occurrence (no confirmed sightings according to Jakubiec & Buchalczyk, 1987) at the edge of the Bieszczady and Beskid Niski, then widens again within the Beskid Niski, showing more but less favourable options for bear movement in areas designated as permanent bear presence and bear dispersal. The main corridor branch narrows again at the edge of the Beskid Niski and Beskid Sądecki, crossing areas of lower probability of bear occurrence, while areas with permanent bear presence (according to Jakubiec & Buchalczyk, 1987) were located more to the north, on the foothills.

In 2013, the designated corridors almost completely fell within the areas of permanent and sporadic bear presence according to Chapron et al. (2014). The corridors were partly wider than in the 1970s, and only in locations indicating sporadic bear occurrence were they divided into multiple, thin branches. By 2013, the share of forest was already very high; however, the forest patches were locally separated by more densely populated villages located in the valleys, which limited connectivity locally. This is especially visible in the western part of the study area, where overall human impact is much higher than in the eastern part, and bear occurrence is rather sporadic (Fig. 4). This is the area that is still critical for connectivity between western and eastern bear habitats (Kaczensky et al., 2021).

This supports the argument that while the overall increase in forest cover is positively correlated with connectivity for large carnivores in the Carpathians, its effectiveness is highly limited by scattered settlements, especially in the valleys (Kaim et al., 2019; Kaim et al., 2024). As valleys provide the most favourable conditions for roads and settlements in mountainous regions, the density of development in these areas may pose a significant threat to the conservation of sustainable dispersal corridors. While the suitable passage in this region is already relatively narrow, further expansion of developed areas traversing this area from north to south may be detrimental to future west-east bear connectivity, which calls for effective spatial planning procedures (Ćwik, 2024).

Our study refers to land cover changes since the mid-19th century, a period during which the minimum forest cover was observed in the region and has increased over time. This pattern is typical for many areas in Europe due to the phenomenon of forest transition (Meyfroidt & Lambin, 2011; Kozak & Szwagrzyk, 2016). The forced displacement of inhabitants accelerated the dynamics of forest cover increase, although the trend was already visible earlier. For forest specialist species, this suggests that habitat conditions have generally improved since the mid-19th century, when they were likely most challenging. At that time, wild animals were often considered pests and actively eradicated (Niezabitowski, 1933; Jakubiec, 2001). Interestingly, a detailed habitat reconstruction of the distribution of wolves in Spain over the last 150 years showed that wolves were more widespread in the mid-19th century than in the 1970s or even today, which contrasts with the situation in central Europe (Clavero et al., 2023). This indicates that more effort is needed to better understand the long-term relationships between species occurrence and land use change. On one hand, environmental reconstructions based on historical data should be used with proper knowledge and caution (Clavero & Revilla, 2014; Clavero et al., 2022). On the other hand, they are critical for defining shifting baselines and policy-relevant recovery targets (Grace et al., 2019).

Conclusions

Our study identified a substantial impact of post-war resettlement and associated land use changes on brown bear habitat connectivity in the Polish Carpathians. The forced displacement of the local population and subsequent land abandonment led to a widespread increase in forest cover, reduced human pressure, and created favourable conditions for brown bear movement and habitat improvement. These changes facilitated the merging of eastern and western Carpathian bear populations over time, which is critical for species recovery. This study demonstrates the consequences of the past, although politically driven, depopulation on the local environmental conditions. The results are particularly important in the context of gradual decline of inhabitants observed in various part of contemporary rural Europe. A better understanding of these processes will aid in shaping future restoration policies from both environmental and societal perspectives.

Supplemental Information

Supplemental Information 1 Code used to create cost surfaces

Supplemental Information 2 A set of least-cost corridors for 1860s, 1930s, 1970s and 2013

Supplemental Information 3 A set of cost surfaces for 1860s, 1930s, 1970s and 2013

Supplemental Information 4 Supplemental tables and figures

The authors thank the referees and the editor for their constructive comments.

Additional Information and Declarations

Competing Interests

Author Contributions

Data Availability

The authors declare there are no competing interests.

Anna Szwagierczak conceived and designed the experiments, performed the experiments, analyzed the data, prepared figures and/or tables, authored or reviewed drafts of the article, and approved the final draft.

El.zbieta Ziółkowska conceived and designed the experiments, authored or reviewed drafts of the article, and approved the final draft.

Joanna Wąs conceived and designed the experiments, prepared figures and/or tables, authored or reviewed drafts of the article, and approved the final draft.

Michał Jakiel conceived and designed the experiments, prepared figures and/or tables, authored or reviewed drafts of the article, and approved the final draft.

Dominik Kaim conceived and designed the experiments, authored or reviewed drafts of the article, and approved the final draft.

The following information was supplied regarding data availability:

The data and code are avaliable at Zenodo:

Szwagierczak, A., Ziółkowska, E., Wąs, J., Jakiel, M., & Kaim, D. (2025). Raw data for the study: Szwagierczak et al. “Impact of historical land use change on the brown bear habitat connectivity in the Polish Carpathians” [Data set]. Zenodo. https://doi.org/10.5281/zenodo.15395647

The data and code are available in the Supplemental Files.

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
