# Peer review of "Impact of historical land use change on the brown bear habitat connectivity in the Polish Carpathians"

_PeerJ, doi:10.7717/peerj.20295_

## Round 0.1 · original submission · Major Revisions

· Academic Editor

Major Revisions

·

Basic reporting

I read the paper it is interesting paper with a novelty in findings I added my comments and suggestions on the file which can be shared by authors

Experimental design

I think the experimental design is fine, only the author needs to explain some parts of the methods and findings, which I commented on in the main document

Validity of the findings

The findings are valid

Additional comments

Please see the attached file

Reviewer 2 ·

Basic reporting

This study represents an attempt by the authors to shed light on the impact of historical land-use changes on potential connectivity of brown bear habitats in the region between the western and eastern parts of the Carpathian Mountains. Specifically, the authors used several digital environmental variables derived for four different time periods and then analysed landscape connectivity through their estimated resistance surfaces and a least-cost path analysis with two different approaches (i.e., weighted and unweighted).



Figure 1: you cannot see where the lighter colour is within the figure itself.

Experimental design

Despite a great deal of analytical/mapping work, the manuscript does not present any new relevant information on the actual connectivity for the study species in that study area and instead has some significant methodological flaws that can easily affect the outcomes. Below, I list the most relevant critical aspects of this work, following the order within the Introduction/Methods text (i.e., not in importance), also offering some comments/suggestions to improve the manuscript itself. There are more suggestions for improving the manuscript in the Results and Discussion sections, but I suggest that the authors address the methodological/conceptual flaws first.

Lines 59-61: what does rewilding have to do with the (natural) recovery of large carnivores in Europe?
Lines 68-70: very unclear sentence, it does not tie in with what the authors previously stated.
Lines 77-80: the concept here is unclear and controversial, first the authors state that greater habitat availability/suitability does not automatically mean greater ecological connectivity (lines 65-66), now they make this assumption by talking about “unintentional improvement in ecological connectivity”.
Lines 97-110: the general aim of the entire work, i.e., the so-called research question/gap, is missing. The authors repeat several times in the introduction that the general habitat suitability for brown bears has increased over time (due to increased forest cover and less anthropogenic activity/disturbance), and that the connectivity between the habitats of the western and eastern portions of the Polish Carpathians has increased. What is the actual rationale of what the authors actually analyse? In other words, why should the analysis of “past” connectivity in that area be relevant today but especially for the future management (cfr. line 107)? If the authors wanted to derive guidelines for the future, given the ongoing climate change (line 49), was it not much more meaningful to analyse connectivity in the present and then project it into the future (Ashrafzadeh et al., 2022), given climate change or the potential increase in human-bear conflict?
Lines 129-145: in the first of the two paragraphs we go back in time to the previous paragraph, while the entire second paragraph here is practically a repetition of concepts already expressed in the introduction.
Lines 159-163: what is not clear to me here is why the authors did not only focus on the important (or potentially important) environmental factors/variables in or around their study area.
Lines 183-189: the authors provide no explanation as to why they chose to use the least-cost path method and not, e.g., circuit theory by Circuitscape, which could have allowed them to analyse and focus on general/regional connectivity in their study area and towards each direction (i.e., “omnidirectional” approach) (Koen et al., 2014).
Lines 196-201: in my opinion, these assumptions made by the authors are the most “problematic” part of the chosen methodology. Here, indeed, the authors make a series of assumptions about the movement behaviour of bears which (apart from the last one), in the absence of supporting data in their study area, can hardly be considered valid and, overall, can influence the results.
For example, regarding the first assumption, why would bears prefer to move around in the innermost parts of forests rather than near forest edges? In fact, interiors/older forests are often associated with behaviour and needs for resting and shelter, but not with more active behaviour (e.g., Ciarniello et al., 2014; Moe et al., 2007; Nielsen et al., 2004). By the way, the study of Ziółkowska et al. (2016) showed that brown bears in the Carpathians travelled closer to the forest edges.
Furthermore, regarding the second assumption, what the authors assume is not always true: brown bears and other large carnivores often use human infrastructures to move and travel, or at least increase their movement rates near them (see e.g., Dickie et al., 2020; Donatelli et al., 2022; Falcinelli et al., 2024; Roever et al., 2010, 2008). Therefore, if movement behaviour is considered (as should be done when performing connectivity models, e.g., Maiorano et al., 2019), travel costs may actually be higher away from human infrastructure.
Furthermore, have these assumptions been valid and consistent over the past 160 years? How can the authors be sure of this, especially if anthropogenic disturbance was generally less significant in the past?
Lines 238-239: based on what has been said above, this method is so conceptually flawed. What is the point of validating resulting movement corridors with bear presence/occurrence data (indicating mostly the habitat suitability)? In general, ecological corridors facilitate animal movements (e.g., dispersal, migration) and are species-specific. These long-distance movements (especially in the case of large carnivores such as bears, and especially if dispersing) do not always reflect the use of safer/suitable areas (i.e., in terms of habitat) but simply areas that favour movement. Thus, roads, paths or trails, and even more disturbed areas can also be used to this scope (see also Barry et al., 2020; Thorsen et al., 2022).

Validity of the findings

No comment

Additional comments

Ultimately, my advice to authors is to use data on the presence of the species in their study area (which can also be downloaded from GBIF, for example) to model current connectivity using the modelling methods available today (e.g., Boudreau et al., 2022; García-Sánchez et al., 2022). Alternatively, authors could use studies already carried out near their study area (Ziółkowska et al., 2016) to create an evidence-based resistance surface and then model connectivity, see Rodrigues et al., (2022).


Literature cited in the review

Ashrafzadeh, M.R., Khosravi, R., Mohammadi, A., Naghipour, A.A., Khoshnamvand, H., Haidarian, M., Penteriani, V., 2022. Modeling climate change impacts on the distribution of an endangered brown bear population in its critical habitat in Iran. Sci. Total Environ. 837, 155753. https://doi.org/10.1016/j.scitotenv.2022.155753
Barry, T., Gurarie, E., Cheraghi, F., Kojola, I., Fagan, W.F., 2020. Does dispersal make the heart grow bolder? Avoidance of anthropogenic habitat elements across wolf life history. Anim. Behav. 166, 219–231. https://doi.org/10.1016/j.anbehav.2020.06.015
Boudreau, M.R., Gantchoff, M.G., Ramirez-Reyes, C., Conlee, L., Belant, J.L., Iglay, R.B., 2022. Using habitat suitability and landscape connectivity in the spatial prioritization of public outreach and management during carnivore recolonization. J. Appl. Ecol. 59, 757–767. https://doi.org/10.1111/1365-2664.14090
Ciarniello, L.M., Heard, D.C., Seip, D.R., 2014. Grizzly bear behaviour in forested, clearcut and non-forested areas in sub-boreal British Columbia. Can. Wildl. Biol. Manag. 3, 82–92.
Dickie, M., McNay, S.R., Sutherland, G.D., Cody, M., Avgar, T., 2020. Corridors or risk? Movement along, and use of, linear features varies predictably among large mammal predator and prey species. J. Anim. Ecol. 89, 623–634. https://doi.org/10.1111/1365-2656.13130
Donatelli, A., Mastrantonio, G., Ciucci, P., 2022. Circadian activity of small brown bear populations living in human-dominated landscapes. Sci. Rep. 12, 15804. https://doi.org/10.1038/s41598-022-20163-1
Falcinelli, D., del Mar Delgado, M., Kojola, I., Heikkinen, S., Lamamy, C., Penteriani, V., 2024. The use of anthropogenic areas helps explain male brown bear movement rates and distance travelled during the mating season. J. Zool. https://doi.org/10.1111/jzo.13199
García-Sánchez, M.P., González-Ávila, S., Solana-Gutiérrez, J., Popa, M., Jurj, R., Ionescu, G., Ionescu, O., Fedorca, M., Fedorca, A., 2022. Sex-specific connectivity modelling for brown bear conservation in the Carpathian Mountains. Landsc. Ecol. 37, 1311–1329. https://doi.org/10.1007/s10980-021-01367-8
Koen, E.L., Bowman, J., Sadowski, C., Walpole, A.A., 2014. Landscape connectivity for wildlife: Development and validation of multispecies linkage maps. Methods Ecol. Evol. 5, 626–633. https://doi.org/10.1111/2041-210X.12197
Maiorano, L., Chiaverini, L., Falco, M., Ciucci, P., 2019. Combining multi-state species distribution models, mortality estimates, and landscape connectivity to model potential species distribution for endangered species in human dominated landscapes. Biol. Conserv. 237, 19–27. https://doi.org/10.1016/j.biocon.2019.06.014
Moe, T.F., Kindberg, J., Jansson, I., Swenson, J.E., 2007. Importance of diel behaviour when studying habitat selection: examples from female Scandinavian brown bears (Ursus arctos). Can. J. Zool. 85, 518–525. https://doi.org/10.1139/Z07-034
Nielsen, S.E., Boyce, M.S., Stenhouse, G.B., 2004. Grizzly bears and forestry I. Selection of clearcuts by grizzly bears in west-central Alberta, Canada. For. Ecol. Manage. 199, 51–65. https://doi.org/10.1016/j.foreco.2004.04.014
Rodrigues, R.G., Srivathsa, A., Vasudev, D., 2022. Dog in the matrix: Envisioning countrywide connectivity conservation for an endangered carnivore. J. Appl. Ecol. 59, 223–237. https://doi.org/10.1111/1365-2664.14048
Roever, C.L., Boyce, M.S., Stenhouse, G.B., 2010. Grizzly bear movements relative to roads: Application of step selection functions. Ecography (Cop.). 33, 1113–1122. https://doi.org/10.1111/j.1600-0587.2010.06077.x
Roever, C.L., Boyce, M.S., Stenhouse, G.B., 2008. Grizzly bears and forestry II: Grizzly bear habitat selection and conflicts with road placement. For. Ecol. Manage. 256, 1262–1269. https://doi.org/10.1016/j.foreco.2008.06.006
Thorsen, N.H., Hansen, J.E., Støen, O.G., Kindberg, J., Zedrosser, A., Frank, S.C., 2022. Movement and habitat selection of a large carnivore in response to human infrastructure differs by life stage. Mov. Ecol. 10, 1–14. https://doi.org/10.1186/s40462-022-00349-y
Ziółkowska, E., Ostapowicz, K., Radeloff, V.C., Kuemmerle, T., Sergiel, A., Zwijacz-Kozica, T., Zięba, F., Śmietana, W., Selva, N., 2016. Assessing differences in connectivity based on habitat versus movement models for brown bears in the Carpathians. Landsc. Ecol. 31, 1863–1882. https://doi.org/10.1007/s10980-016-0368-8


Reviewer 3 ·

Basic reporting

Abstract
Line 30: Preference refers to what an animal would choose in an ideal scenario with no limitations. It is virtually impossible to measure “preference”, given the many drivers (availability, physiology, season, …) leading to it. See for example page 6 here for an example: https://www.carnivoreconservation.org/files/thesis/basille_2008_phd.pdf. In this case, and throughout the article, refer to “USE” instead, or “selection” if appropriate.
Lines 28–31: In “We used two different approaches to create a cost surfaces…” drop the ‘a’ as surfaces is plural (“...two different approaches to create cost surfaces”)

Introduction
Line 77-80: this concept is already broadly touched on Lines 65-67 already, you could consider removing it as somehow redundant.
Line 91 (and throughout the text): Use the term “dispersing individuals” rather than “migratory”, as the latter can be misleading when referring to animals (migration in ecology- or ethology- has a different meaning; Dorst, 2019, Migration, Encyclopædia Britannica, inc., retrieved from www.britannica.com/science/migration-animal). See also e.g. Zedrosser, Andreas, et al. "Should I stay or should I go? Natal dispersal in the brown bear." Animal Behaviour 74.3 (2007): 369-376.

Material and Methods
Line: 121-152: This section in my opinion, while very relevant, does not belong to the Materials & Methods, but rather to the introduction. The forced resettlement of the Lemko population in the 1940s and its ecological consequences are essential to understanding why habitat connectivity changed. This information should be provided earlier in the introduction, not in the Materials & Methods section, which should provide more technical details.
Line 143: “further land abandonment is visible”. What does it mean? Is it still ongoing?
Line 151: “which are considered areas of permanent bear presence and between 152 which connectivity was analysed”. Please cite the last EU LCIE report: (Kaczensky, P., Ranc, N., Hatlauf, J., Payne, J.C. et al. 2024. Large carnivore distribution maps and population updates 2017 – 2022/23. Report to the European Commission under contract N° 09.0201/2023/907799/SER/ENV.D.3 “Support for Coexistence with Large Carnivores”, “B.4 Update of the distribution maps”. IUCN/SSC Large Carnivore Initiative for Europe (LCIE) and Istituto di Ecologia Applicata (IEA).)
Lines 213–215: I disagree with this statement. In the past, humans were far ubiquitous in the landscape and used the land much more intensively than they do today. While modern urbanization and tourism has increased localized disturbance and noise, historic activities like herding and agriculture were pervasive and widespread throughout the mountains, and hunting was also legal.

Discussion
Line 297-319: this part is not introducing entirely new concepts, but rather re‐states and interprets material already laid out in the Introduction, placing it in the context of your results. I suggest re-writing it more tightly because it feels redundant with the other information reported in the introduction.
Line 358-366: this section needs to be backed up by literature. Distance‐based and density‐based metrics capture fundamentally different dimensions of human presence and are appropriate at different spatial scales. A single building may drive a distance‐to‐feature layer yet impose negligible disturbance on brown bears, whereas density metrics reflect aggregate human pressure. Although proximity (noise, visual detection, encounters) is ecologically relevant, it isn’t clear that distance is always better than density without citing studies.
Line 369-371: den sites areas and ecological corridors are not the same thing. Den sites are refuge or resting areas, whereas ecological corridors are the routes bears use to travel between habitat patches. These serve fundamentally different functions–corridors facilitate movement, while den sites provide shelter. I suggest rephrasing this past.

Figures and Tables
Figure 2, 3: Color palettes may not be distinguishable for readers with color-vision deficiencies. Use color-blind–friendly palettes and include distinct line styles or symbols in the legend.

Experimental design

My main concern with the study is the rationale for—and implementation of—an expert-based approach to derive connectivity corridors. Although I understand that the authors lack access to empirical movement data, the manuscript provides insufficient detail on how literature‐derived effect sizes were translated into the numeric scores in the cost_values.txt file attached. The cited studies (Lines 161-163) span different regions, seasons, and ecological questions (from den‐site selection to broad‐scale habitat use), so it is unclear how their β-coefficients or odds ratios became the cost values applied here. A transparent protocol—such as extracting each paper’s effect sizes, normalizing them, and then mapping those normalized weights to the scores provided—would greatly improve reproducibility. Likewise, involving regional brown bear experts in a structured way (for example, using Delphi rounds or pairwise comparisons and calculating inter-rater agreement like Cohen’s κ) would help ensure that the assigned scores truly reflect local conditions. While expert-based methods can be limited (e.g. Zeller, Katherine A., Kevin McGarigal, and Andrew R. Whiteley. "Estimating landscape resistance to movement: a review." Landscape ecology 27 (2012): 777-797.), a more standardized and documented weighting procedure would nonetheless yield ‘more defensible’ cost estimates. Finally, if presence–absence or presence-only data ever become available for the study area, fitting a simple resource‐selection function or MaxEnt model to empirically derive resistance weights could dramatically improve the cost-surface analysis. I recognize that data access may be constrained, but even a basic empirical calibration would substantially strengthen the study’s conclusions.

Validity of the findings

See the comments above regarding the experimental design. I could provide a more robust comment on the validity of the findings with a revised methodology.

Additional comments

In my opinion the manuscript addresses a relevant and regionally significant issue by examining changes in brown bear habitats over an extensive time period (1860s–2013) and the resulting connectivity corridors. The authors leverage historical data in an unprecedented way to provide insight into the effects of socio-ecological changes on the presence and persistence of bears. While the manuscript is conceptually interesting, it must be improved analytically before it can be considered for publication.

---

## Round 0.2 · accepted · Accept

· Academic Editor

Accept

This revised version is suitable for publication in PeerJ.

·

Basic reporting

-

Experimental design

-

Validity of the findings

-

Reviewer 3 ·

Basic reporting

-

Experimental design

-

Validity of the findings

-